# Alginate Oligosaccharides Affect Mechanical Properties and Antifungal Activity of Alginate Buccal Films with Posaconazole

**DOI:** 10.3390/md17120692

**Published:** 2019-12-09

**Authors:** Marta Szekalska, Magdalena Wróblewska, Monika Trofimiuk, Anna Basa, Katarzyna Winnicka

**Affiliations:** 1Department of Pharmaceutical Technology, Medical University of Białystok, Mickiewicza 2C, 15-222 Białystok, Poland; magdalena.wroblewska@umb.edu.pl (M.W.); monikam@umb.edu.pl (M.T.); 2Institute of Chemistry, University of Białystok, Ciołkowskiego 1K, 15-245 Białystok, Poland; abasa@uwb.edu.pl

**Keywords:** alginate, alginate oligosaccharides, posaconazole, buccal films, antifungal activity, freeze-thaw technique

## Abstract

Sodium alginate and its oligosaccharides through potential antifungal properties might improve the activity of antifungal drugs enhancing their efficacy and potentially reducing the frequency of application. Mucoadhesive buccal films are oral dosage forms designed for maintaining both local or systemic drug effects and seem to be a very promising alternative to conventional oral formulations. Hence, in this study, mucoadhesive buccal films based on the alginate and its oligosaccharide oligomer composed predominantly of mannuronic acid for the administration of posaconazole-antifungal drug from the azole group were developed. As the polymer gelation method, a relatively new freeze-thaw technique was chosen. All prepared formulations were examined for pharmaceutical tests, swelling, mechanical, and mucoadhesive properties. In addition, the influence of sodium alginate (ALG) and alginate oligosaccharides (OLG) on POS antifungal activity on *Candida* species was performed. It was observed that film formulation containing 1% ALG and 1% OLG (F2) was characterized by optimal mucoadhesive and swelling properties and prolonged drug release up to 5 h. Additionally, it was shown that OLG affected the growth reduction of all tested *Candida spp*. The obtained data has opened the way for future research for developing OLG-based dosage forms, which might increase the activity of antifungal drugs.

## 1. Introduction

Fungal infections, despite the constant progress of medicine, still constitute a valid therapeutic problem. One of the most common opportunistic fungal infections is candidiasis, which is generally caused by *Candida* species. *Candida* is a genus of yeast belonging to the normal, commensal microbiota of the gastrointestinal tract, vagina, skin, oral cavity, or other mucosal surfaces in healthy humans. In the case of patients critically ill, with immunodeficiency or AIDS, patients that use total parenteral nutrition after transplantation and treated by cytotoxic chemotherapies or broad-spectrum antibiotics, *Candida spp.* might contribute to systemic infection. Although the genus *Candida* consists of various species, predominant etiological agents of human infection are *Candida albicans*, *Candida parapsilosis*, *Candida tropicalis,* and *Candida krusei*. In addition, *Candida albicans* causes oral candidiasis – one of the most common opportunistic fungal infections, where local drug dosage forms are effective, especially buccal films. While a wide range of available antifungal substances can be used for effective treatment of local candidiasis, in the prevention of invasive candidiasis, systemic antifungal chemotherapy is recommended. The first-line drugs used to treat candidiasis include echinocandins and azoles [1].

Posaconazole (POS) is a new triazole antifungal agent with a broad spectrum of antifungal activity against commonly encountered fungal species, such as *Candida spp., Cryptococcus neoformans, Aspergillus spp., Zygomycetes*, as well as endemic fungi and dermatophytes. The mechanism of POS action is based on the biosynthesis inhibition of ergosterol, the main sterol component of fungal cytoplasmic membranes. However, POS is characterized by low solubility in an aqueous and acidic environment, and its absorption is limited by the dose, depending on the food, which, in consequence, affects the low bioavailability [2].

Buccal films are mucoadhesive oral dosage forms that consist of a hydrophilic polymer, active substance, and other excipients. Films are applied on the inner membrane of the buccal mucosa, where after contact with the saliva, they hydrate, adhere, and dissolve drug that results in local or systemic effects. Due to the high flexibility, the comfort of application, precise dosage, and a larger area of drug absorption, they are promising alternatives to conventional oral formulations. They have been shown to reduce the toxicity caused by conventional drug therapies, to improve drug efficacy and bioavailability, and to reduce the dose and frequency of administration. They are designed as oral delivery systems for compounds with unfavorable pharmacokinetic properties [3].

Alginates constitute a family of natural polysaccharide polymers extracted from brown seaweeds bacteria or produced by the bacteria species: *Azotobacter* and *Pseudomonas*. Chemically, they represent linear copolymers consisting of β-D-mannuronate (M) and α-L-guluronate (G) residues linked with α- and β-1,4-glycosidic bonds. The M and G residuals are linked together in different sequences – heteropolymeric (MG) and/or homopolymeric (M or G) blocks, which determine the properties of the polymer. The group of alginates includes sodium alginate (ALG)—sodium salt of alginic acid with unique swelling, gelling, and mucoadhesive properties [4,5]. Due to their biological activity, great attention is drawn to the alginate oligosaccharides (OLG)—a depolymerized product of alginate by alginate lyase [6]. It has been reported that they possess antioxidant, anti-inflammatory, and antibacterial properties. Additionally, OLG inhibits fungal cell growth and potentiates the activity of antifungal drugs against *Candida*, which suggests their potential role in the treatment of fungal infections [7,8]. It has been observed that ALG improves the bioavailability of imidazoles and triazoles and ALG-based antifungal drug delivery systems, provided sustained drug release, leading to a reduction in a drug dose, the administration frequency, and drug toxicity [9].

In spite of the fact that OLG has been shown to be a promising agent for antifungal therapy, there has not been yet any attempts to evaluate them as carriers for drug delivery. Therefore, in the present study, the impact of ALG and its OLG (composed predominantly of mannuronic acid) on POS antifungal activity was tested. As the polymer gelation method, freeze-thawing was utilized. Compared with conventional chemical cross-linking, it is a relatively new physical technique, with no influence on the polymer biocompatibility, biodegradability, and toxicity. This technique is based on three phases: freezing of a monomeric or polymeric solution or zol, its storage in the frozen state, and thawing. Solvent crystallization during freezing might lead the polymer chains to connect into structures with a large number of macropores. ALG gels obtained using freeze-thaw method demonstrate a significant shrinkage that indicates a high degree of a balance between repulsive and attractive interactions, which, as a consequence, might lead to the creation of junction zones through Van der Waals forces and between uncharged ALG hydrogen bonds chains in the unfrozen liquid phase during thawing [10,11]. In our work, the OLG effect on the pharmaceutical characteristics of ALG buccal films was also examined. All prepared formulations were tested for mechanical properties, drug content uniformity, and the in vitro POS-release. As mucoadhesive properties are crucial for buccal formulations, ex vivo and in vivo tests were additionally performed. Physicochemical properties of prepared films were examined by scanning electron microscopy (SEM) and differential scanning calorimetry (DSC), and ALG and OLG impact on POS antifungal activity on the *Candida* species was also assessed.

## 2. Results and Discussion

Ionic cross-linking is the most widely used method of ALG gelation, but it is fraught with many disadvantages, such as difficulties in controlling the gelation rate, which affects structure uniformity, the deterioration gels mechanical properties, and limited long-term stability [6]. One of the relatively new methods to prepare ALG gels is the freeze-thaw technique, also known as cryogelation. This method involves solvent crystallization during freezing, which leads to the reduction of the space of polymer chains, increases the polymer concentrations and forces between polymer chains, leading to their connection and formation of the hydrogels after thawing. Gels created under such conditions are known as cryogels. Cryogelation is a complex process and consists of three steps: non-deep freezing of a polymer solution, its storage in the frozen state, and thawing. The obtained cryogels are characterized by elasticity and a soft, porous structure with high permeability for drug molecules [11,12]. Tentative tests comparing the effect of solvent-casting and freeze-thaw method on placebo film characteristics showed that films formulated by freeze-thawing possessed significantly better mechanical, swelling, and mucoadhesive properties (data presented in Appendix A). A schematic structure of ALG cryogels preparation by the freeze-thaw method is presented in Figure 1.

After preliminary studies, to obtain films with appropriate properties, 1% and 2% ALG solutions with beneficial viscosity were chosen. The too low viscosity of solutions led to very thin films with poor mechanical properties, but too high prevented the pouring solution on the plexiglass molds. Viscosity values of prepared ALG/OLG solutions P1_sol_–P4_sol_ (measured before the freeze-thaw process) and cryogels P1_gel_–P4_gel_ are presented in Table 1. The obtained data indicated that the cryogelation process significantly increased the viscosity of ALG/OLG gels. The obtained cryogels were used to prepare films. The composition of film formulations placebo (P1–P4) and POS-loaded (F1–F4) is given in Table 2.

### 2.1. Characteristics of Buccal Films

The effect of freeze-thaw gelation on the surface and cross-section of ALG and OLG film structure is shown in Figure 2 and Figure 3. The surface of ALG films placebo (formulation P1) without OLG addition was smooth and homogenous with some irregularities (Figure 2a, Figure 3a). Films P4 and F4, consisting of only OLG, generated a heterogeneous surface with gelled areas and porous presence (Figure 2b,d). The cross-section of the OLG films showed a sponge-like structure with lots of pores in the matrix (Figure 3b). SEM micrographs of surface and cross-section of POS-loaded films exhibited non-homogenous nature and the presence of drug crystals (Figure 3c,d). In addition, the formulation of the film, with POS (Figure 2c,d and Figure 3c,d), possessed a crystalline structure, and the original polymer network was invisible.

It was observed that the films with placebo were soft, dry, tack-free, and transparent with the slightly yellow coloration of ALG. On the contrary, the formulations of POS-loaded films were white. In all prepared films, the moisture content was in the range from 5.3 ± 1.5% to 9.7 ± 0.2%. The pharmaceutical characteristics of the designed buccal films are presented in Table 3.

Film thickness is an important factor affecting mechanical and swelling properties and drug release. It was observed that the thickness of films increased significantly (p<0.05) with the increase in ALG concentration and POS presence. However, formulations containing only OLG possessed lower thickness. The prepared films did not show any significant differences in the thickness within the formulation, indicating the good repeatability of the process. However, the differences in the thickness between formulations placebo and POS-loaded were noted (Table 3).

Drug content uniformity is an important quality control parameter, which should be in the range from 85% to 115% [13]. It was perceived that POS content in all formulations was comparable (95.4 ± 0.1%–103.8 ± 6.2%) and within the range of pharmacopoeial requirements, suggesting that the polymer type and its concentration did not influence the uniformity of drug under the current experimental condition (Table 3). As non-physiological pH might irritate the buccal mucosa [14], hence the surface pH of prepared films was measured. The prepared films exhibited pH in the range of 6–7, indicating compatibility with buccal pH.

Disintegration time is a crucial feature affecting drug release and depends on the medium volume; therefore, it was determined in the pharmacopoeial apparatus and on the Petri dish with medium volume imitating the volume of the saliva in the human mouth. It was concluded that disintegration time was increased with the increasing polymer concentration, film thickness, POS presence, and with the reduced medium volume. When the 7 mL of medium was used, extended disintegration time was observed, which is a beneficial feature for buccal formulation. Prolonged disintegration provides increased adhesion to the mucosa, and consequently, longer drug release time [15].

### 2.2. Mechanical Properties

Mechanical properties influence the packaging, transport, or storage condition and application of films by patients. The mechanical features of designed films were examined by four different parameters: tensile strength (TS), percent of elongation (E%), Young’s modulus (E), and folding endurance (Figure 4). For mucoadhesive and buccal drug administration, flexible and strong films are preferable, and they should be characterized by the low value of Young’s modulus and high value of folding endurance, tensile strength, and percent of elongation [16]. Obtained data showed that formulations prepared only from ALG (P1 and F1) possessed the lowest TS values and relatively high E% and low E values (Figure 4a–c). However, films prepared only from OLG (formulations P4 and F4) showed the maximum percent of elongation among the formulations, relatively high TS, and the lowest E values (Figure 4a–c). In the case of formulations P2 and F2, OLG addition improved mechanical properties, expressed as the increase in values of TS and percent of elongation and the decrease in Young’s modulus compared to the formulation prepared only with ALG. On the contrary, formulations P3 and F3 were characterized by the highest TS and E but the lowest E%. The obtained results suggested that these formulations were rigid, which was caused by the presence of a higher concentration of ALG and OLG in the films and higher film thickness. OLG presence in the formulations led to obtaining more soft and flexible films. This behavior was in agreement with data reported by Costa et al., who indicated that the ALG mechanical properties were M/G (mannuronic acid/guluronic acid) ratio-dependent, and ALG with higher M-blocks was characterized by higher TS and E% than ALG with dominated G bocks [17]. In addition, it was observed that placebo formulations (P1–P4) were characterized by higher TS, E%, and E values than POS-loaded formulations (F1–F4), proving that the addition of the drug substance weakens mechanical properties.

Folding endurance was measured to determine the rupturing resistance of designed buccal films, and, in all formulations, it was over 100 times, exhibiting good flexibility [18].

The mechanical properties are affected by the chemical composition of ALG—an increased amount of guluronic (G) block units in polymer chains forms stiffer, brittle, and mechanically more stable gels. On the contrary, ALG, characterized by high values of mannuronic (M) blocks, creates soft and more elastic gels. However, MG-blocks in ALG gel determine its shrinkage and higher flexibility [19]. In the study performed by Ashikin et al. [19] and Azeredo et al. [20], it was confirmed that mechanical properties were ALG chemical composition dependent. They reported that ALG gels with high G-blocks presence were significantly less flexible than gels with high M-blocks. In addition, Liang et al. proved that the freeze-thaw process impacted positively on tensile strength and elongation at break, but simultaneously affected Young’s modulus [21]. It was also shown that methods based on the solvent evaporate (freeze-thaw method and solvent-casting) improved the mechanical properties of the polymer, as solvent remained in the films to a small extent and performed the plasticizer function [22].

### 2.3. Swelling Study

Swelling is the first step of the formation of adhesive bonds between polymer and mucosa, affecting the residence time at the application site and, as a consequence, the drug release and its increase absorption through buccal mucosa [23]. After swelling, the polymer chains are extending, exposing the binding sites, and the adhesion between polymer and mucosa occurs, but initially, the bond formed is weak. Mucoadhesion extends during polymer hydration until the moment in which it leads to decrease adhesive strength in consequence of disentanglement polymer chains with tissue [24].

In this study, swelling characteristics of films were examined in the function of time. As the swelling media, the distilled water and simulated saliva solution (SSS) were applied. The observed swelling index (SI) of developed formulations is depicted in Figure 5. Prepared films exhibited different swelling properties depending on the medium. Formulations P1–P3 showed higher SI in the range from 27.25 ± 0.9 to 14.5 ± 0.65 at 120 min when SSS was used (Figure 5a). It was observed that SI was ALG concentration-dependent – formulation P3 with high ALG concentration possessed higher SI properties. Similarly, Sarheed et al. noted that swelling ability was increased with the increased ALG concentration in the film [25]. Interestingly, formulation P4, consisting of only OLG, represented high but short swelling ability only up to 60 min with SI value 15.92 ± 1.91. Films prepared only from OLG showed decreased swelling in the SSS compared to the pure ALG. Reduction of films’ water uptake in the formulations P2 and P3 in comparison with the formulation F1 might be related to the OLG concentration increase in the films. Short chains of OLG oligomers present in the film matrix diffuse into the medium, thereby reducing osmotic pressure. It results in the relaxation of polymer chains and, as a consequence, the SI increase. When the film structure is fully hydrated, it begins to lose its structural integrity due to the medium penetration into the matrix, and ALG chains begin to disintegrate and dissolve. Prolonged swelling of ALG formulations is related to a high number of G-blocks, which form an integral matrix slowly dissolving in SSS and possessing the ability to slower water uptake. In addition, the presence of Na+ ions in the medium leads to electrostatic repulsion and, in consequence, causes the chain disintegration. In the gel matrix, free counter-ions form an ‘ionic’ osmotic pressure between solvent and gel, which provides slow solvent diffusion into the film polymer network [26].

Furthermore, when the swelling study was performed using distilled water, the significantly lower swelling ability of all formulations was observed (Figure 5b). Such a phenomenon was previously observed in the case of non-cross-linking ALG-based films by Costa et al. [17]. In POS-loaded films, a significant decrease in swelling profile was noted. This fact might be related to the poor solubility of POS in SSS and water, hindering the water uptake (Figure 5a,b) [17].

### 2.4. Mucoadhesive Properties

The application of polymers with mucoadhesive properties as a result of direct contact of the polymer with the mucous membrane might significantly extend the contact time of the formulation with the application place. Simultaneously, it might lead to long-term drug effects and, compared to traditional forms, might increase the drug concentration in plasma and improve drug bioavailability. This mechanism is complex and is a result of the occurrence of ionic, hydrogen, covalent bonds, or van der Waals forces [23,27]. The mucoadhesion process may be divided into two stages. In the first step, the polymer in contact with moisturized mucosa becomes wetted, and then polymer chains penetrate through the mucin present in the mucous membrane. In the second phase, the connection is tightened through the creation of the above-mentioned bond between mucin and polymer [27].

During designing buccal films, an important parameter is a beneficial interaction with the mucosa, which maintains film at the site of administration. Bovine buccal mucosa, due to its similar histological epithelium structure to humans [28], implies the usefulness of this model to imitate the in vivo conditions.

The obtained data suggested that ALG and OLG presence affected the mucoadhesive strength of films (Figure 6). Interestingly, it was observed that higher F_max_ values possessed formulations P1 and F1 composed only of ALG (without OLG addition), but higher values of W_ad_ possessed formulation prepared only from OLG (P4 and F4). This fact might be related to M/G-blocks distribution in ALG structure, their conformation in the polymers, and their ability to bind to the buccal mucous membrane. Lower adhesion force of OLG formulations might be a result of high water absorption, contributing to the formation of weaker bonds between OLG and the mucous membrane. In addition, Menchicchi et al. demonstrated that polymer mucoadhesion force was closely related to its molecular weight [29]. It was indicated that ALG chains with low molecular weight remained relatively rigid and, in consequence, less prone to creating bonds with mucin, resulting in lower bioadhesion, than ALG chains with high molecular weight. Furthermore, a higher amount of G-block residues in ALG formed more stable and more swelling gels, which might penetrate the mucosa and create stronger adhesive bonds, exhibiting higher values of F_max_. Placebo films possessed higher values of both F_max_ and W_ad_ than POS-loaded formulations, indicating that active substance hindered the polymers’ contact with the mucous membrane.

Ex vivo retention time evaluated using bovine buccal mucosa is presented in Table 4. The highest retention time was observed in films F1 with a value of 205.00 ± 10.00 min. Formulations F2 and F3 exhibited lower (from 180.67 ± 14.98 for F3 to 111.33 ± 4.04 for F4) mucoadhesion time. The possible explanation could be that in the case of formulation F1, high ALG concentration and film’s thickness retarded film dissolution, which, in turn, increased the film’s retention time. Additionally, POS presence increased film’s thickness and its poor solubility in the medium, impeding the inflow of water to the matrix and prolonging its retention time.

In vivo mucoadhesive studies in human volunteers performed using placebo films showed that the formulation F1, containing only ALG, was characterized by significantly higher residence time than formulation P4, composed of only OLG (Table 4). It was observed that the films’ residence time was affected by the type and the concentration of the polymer. As ALG content in the formulation increased (formulation F3), the significantly (p<0.05) higher mucoadhesive properties were observed. This behavior was in agreement with the results published by Çelik [30], who indicated that the increase in ALG concentration resulted in higher mucoadhesion. The retention time evaluated in ex vivo and in vivo tests was found to be moderately low in film formulation P4, composed of only OLG (Table 4), which might be a result of low molecular weight, low viscosity, and low swelling ability of OLG [17]. Interestingly, when both ALG and OLG were used in higher concentrations (formulation P3), the highest in vivo residence time was noted (about 153 min).

### 2.5. In Vitro POS-Release

As shown in Figure 7, the POS-release profile was followed for 5 h, and POS dissolved gradually without the burst effect. After 5 h, the total amount of POS released from films with ALG (F1), ALG/OLG (F2 and F3), and OLG (F4) was found to be 98.4 ± 7.1%, 100.9 ± 7.3%, 102.6 ± 7.2%, and 101.6 ± 11.6%, respectively. However, the amount of POS released from formulation composed of only OLG (F4) was 100% just after 3 h of the study. The drug release rate is an important factor influencing the efficacy of antifungal compounds as it affects inhibition of the microbes’ growth and prevents drug resistance [31]. Surprisingly, the amount of POS released from formulations F2 and F3 after 3 h was similar, indicating that it was not ALG and OLG concentration-dependent. Formulations F1–F3 possessed a slightly lower POS-release profile compared to formulation F4—composed of only OLG. POS-release test was performed in SSS as a dissolution medium (pH 6.8) with physiological pH. At pH close to 7, the carboxylic groups of alginate ionize and become –COO– in form, which results in weakening the hydrogen bonding in the polymer chain and electrostatic repulsion from –COO− and higher swelling capacity [32,33]. Formulation F4, composed of only OLG, exhibited definitely higher POS-release, which was related to its swelling properties as they determined the solvent penetration into the matrix. Higher ALG content in the formulations increased the thickness of the film’s diffusion layer, obtaining higher swelling ability and prolonged drug dissolution. ALG’s ability to reduce drug diffusion through various polymeric material was also confirmed by Sarheed et al. [25], Skulason et al. [34], and Rana et al. [35].

The POS-release mechanism was determined by fitting the obtained data into various kinetic models (Table 5). It was shown that POS was released according to zero-order kinetics, where the highest curve linearity was observed. The release of POS followed the Higuchi pattern. According to the Korsmeyer–Peppas equation, values of parameter n were from 0.47 to 0.80 and indicated that the mechanism of POS-release was a combination of diffusion and zero-order kinetics. However, a good linear relationship with a regression index from 0.80 to 0.94 in the Hixson–Crowell exemplar might suggest that mechanism of POS-release was based on the diffusion combined with the erosion of films, which is the result of the gradual dissolution of the film in the medium [36]. It is in agreement with results published by Sarheed [25], who studied metronidazole release from ALG hydrogels obtained by the freeze-thaw method.

### 2.6. Differential Scanning Calorimetry (DSC)

DSC is a frequently applied technique in pharmaceutical analysis because of its ability to deliver detailed information about physical and energetic properties of substances, about the presence of impurities and potential interactions in drug formulations [37]. DSC curves of OLG exhibited broad endothermal peaks between 100 °C and 150 °C (Figure 8), which indicated the loss of water content in the polymer. Additionally, sharp exothermic peaks related to ALG decomposition at 248 °C and for OLG at 263.81 °C were observed, which might be due to their decomposition [38]. The melting points of the formulations were characterized by the shift in the temperature and changing the shape of the peaks towards a lower value, which might be due to the freeze-thaw process. POS thermogram exhibited two sharp endothermic peaks, at temperatures 138.73 °C and 172.70 °C, which corresponded to the loss of crystal water and intrinsic melting points of pure POS, respectively (Figure 9) [39]. On the other hand, the POS peak demonstrated a slight decrease in the melting temperature in the formulations F1, F2, and F4 (167.20 °C, 170.52 °C, and 169.50 °C, respectively) in comparison to pure POS (172.70 °C), which was probably a result of interactions between drug and the polymer and the formation of a polymer matrix [40]. In the POS presence, the segmental mobility of the polymer chains becomes restricted, which can change the kinetics of creating a cryogel, and the shift peaks of the decomposition temperature of the polymer were observed [41].

### 2.7. Antifungal Activity

While ALG is used as a drug carrier, increasing their bioavailability and reducing toxicity [16], OLG—characterized by much lower molecular weights—are extensively studied mainly in terms of biological and microbial activity. Pritchard et al. reported that ALG oligosaccharide (Oligo G), with the high content of guluronic acid (>85%), possessed antibacterial and anti-biofilm properties, enhancing the activity of selected antibiotics [7,42]. In addition, Tøndervik et al. proved that Oligo G was characterized by the ability to perturb fungal growth and to increase the activity of conventional antifungal agents [8], which might be particularly important in developing both new drugs and novel drug delivery systems.

In this work, for the first time, the antifungal activity of ALG/OLG placebo and POS-loaded film formulations on *Candida spp.* was explored. According to CLSI guidelines, antifungal activity was evaluated after 48 h of incubation [43]. Obtained results were expressed as the mean of growth inhibition zone diameter (mm) (Figure 10). It was observed that all the formulations, as a consequence of the gradual swelling and POS-release, inhibited the growth of *Candida spp*. It was observed that formulation composed of only OLG (F4) possessed the higher inhibition zone in *Candida albicans* and *Candida parapsilosis* (32.6 ± 4.8 mm and 43.0 ± 1.6 mm, respectively) (Figure 10a,c). When *Candida krusei* was tested, it was comparable with the formulation composed of only ALG (F1) (Figure 10b). Formulation composed of ALG and OLG (F2) was noticed to exhibit stronger effect in comparison to formulation F3 with higher concentrations of both polymers. This fact is related to the lower thickness of formulation F2 and, in consequence, its lower viscosity and greater drug permeability from the swollen matrix. The highest values of the inhibition zone of POS-loaded films after 48 h were observed against *Candida parapsilosis* (from 38.3 ± 1.2 mm in F1 to 43.0 ± 1.6 mm in F4). These results proved that POS could be incorporated in ALG films and subsequently released, thereby inhibiting target microorganisms.

Tøndervik et al. indicated the potentiating effect of oligosaccharides Oligo G consisting mainly of guluronic acid with nystatin, amphotericin B, fluconazole, miconazole, voriconazole, or terbinafine when *Aspergillus spp.* was used [8]. The data obtained in our study suggested that both powders and films composed of ALG—high molecular weight polymer—and OLG—low molecular weight oligosaccharides—with the predominant mannuronic acid content, also possessed the antifungal activity and might be successfully used as a POS carrier increasing drug activity. Furthermore, it was shown that placebo formulations (P1–P4) were characterized by relatively high inhibition zones, and some differences in antimicrobial activity against examined strains were noted. It was shown that both ALG and OLG were characterized by antifungal activity against all examined *Candida* species. The highest zone inhibition was observed in *Candida parapsilosis* (the range between 35.8 ± 1.9 mm to 24.3 ± 4.7 mm) and the lowest values in *Candida albicans* (from 15.9 ± 2.9 mm to 23.7 ± 2.9 mm, respectively). In addition, it was noticed that OLG presence in the formulations F2 and F3 improved their antifungal activity against *Candida albicans* and *Candida parapsilosis*.

The molecular mechanism of ALG and OLG antimicrobial activity is still not explained [44]. However, it is known that negatively charged polymers interact with the outer surface of the fungal cell, causing it to rupture and leak the intracellular content. In addition, the polymer can form a sticky layer on the microbial cell, thus preventing the transport of nutrients by limiting the function of the cell membrane. The antifungal activity might be also associated with the ability of ALG and OLG to chelate ions, which leads to a reduction in the production of metal-dependent proteins with the important biological role, disturbing the microbes homeostasis, blocking their nutrition, and, as a consequence, limiting cell growth and development. In addition, higher antifungal activity of OLG can be associated with the formation of a more elastic gel with lower viscosity, which diffuses more easily through the medium showing a greater zone of growth inhibition of *Candida* cells [5,45].

## 3. Materials and Methods

### 3.1. Materials

Posaconazole (POS) was provided by Kerui Biotechnology Co. LTD (Xi’an, China). Low molecular weight alginate oligosaccharides (OLG), obtained from *Laminaria japonica,* with viscosity 230 mPa∙s for 1% solution at 25 °C (48% mannuronic acid (M), 16% guluronic acid (G), and 36% mixture M/G, molecular weight 2.9 × 10^3^ DA) was provided from Xi’an Haoxuan Biotech Co., Ltd. (CAS – 94035-02-6, Xi’an, China). Sodium alginate (ALG), obtained from *Macrocystis pyrifera,* with medium viscosity (2415 mPa∙s for 1% solution at 25 °C, 61% mannuronic acid (M) and 39% guluronic acid (G), molecular weight 3.5 × 10^5^ Da, M/G ratio of 1.56) and dimethyl sulfoxide were purchased from Sigma Aldrich (CAS – 9005-38-3, Steinheim, Germany). Sodium chloride, potassium phosphate monobasic, sodium phosphate dibasic, glycerol 86%, and sodium dodecyl sulfate (SDS) were from Chempur (Piekary Śląskie, Poland). Methanol and acetonitrile were of HPLC grade and were purchased from Merck (Darmstadt, Germany). Water for HPLC was distilled and passed through a reverse osmosis system Milli-Q Reagent Water System (Billerica, MA, USA). All other ingredients used were of analytical grade. Bovine buccal mucosa was obtained from a local slaughterhouse. The stock cultures of *Candida albicans* ATCC^®^ 10231, *Candida parapsilosis* ATCC^®^ 22019, and *Candida krusei* ATCC^®^ 6528, from American Type Culture Collection, and Sabouraud dextrose agar (SDA) were provided by Biomaxima (Lublin, Poland). Cellulose acetate membrane filters (0.45 µm) were received from Millipore (Billerica, MA, USA). Simulated saliva solution (SSS, pH 6.8) was prepared by dissolving 8 g sodium chloride, 0.19 g potassium phosphate monobasic, and 2.38 g of sodium phosphate dibasic in 1 L of water [46].

### 3.2. Films Preparations

#### 3.2.1. Placebo Films Preparation Using Cryogels Obtained by the Freeze-Thaw Method

Briefly, ALG solutions were obtained by dissolving proper amount of polymer in water (1% *w/w* or 2% *w/w*) with 0.6% (*w/w)* glycerol (used as plasticizer), and they were stirred using mechanical stirrer model DT 200 (Witko, Łódź, Poland) until homogenous mixture appeared. Then, into the polymer solution, OLG (1% *w/w* or 2% *w/w*) was added with continuous stirring. Formulation P1 was prepared by the addition of 0.6% (*w/w)* glycerol to the 2% (*w/w)* ALG solution, and the formulation P4 by the addition of 0.6% (*w/w)* glycerol to the 2% (*w/w)* OLG solution. The composition of placebo formulations (P1–P4) is given in Table 2. Obtained solutions were frozen at −20 °C for 18 h, followed by thawing at 25 ± 1 °C for 6 h [25]. The freeze-thaw cycle was carried out three times. The obtained cryogels were poured into plexiglass molds with the surface 14 × 14 cm and dried at 37 ± 1 °C for 24 h. After drying, films were cut into pieces of 2 × 3 cm.

#### 3.2.2. Solutions and Cryogels Viscosity Measurements

The viscosity of prepared solutions and cryogels was measured using a digital rotational Brookfield DV-III ULTRA Viscometer (Stuttgart, Germany). All measurements of apparent viscosity were carried out in a temperature-controlled environment at 25 ± 1 °C. Solutions and cryogels (0.5 g) were placed in the sampler holder of the viscometer, and the CP-52 spindle (with the shear rate of 2 s^−1^) was lowered into the sample [47].

#### 3.2.3. POS-Loaded Films Preparation Using Cryogels Obtained by the Freeze-Thaw Method

The proper amount of POS was micronized in a mortar for 20 min and then mixed with the ALG, OLG, or ALG/OLG solutions (described in the point 3.2.1.). The composition of the placebo (P1–P4) and POS-loaded (F1–F4) formulations is given in Table 2. Obtained mixtures were frozen at −20 °C for 18 h, followed by thawing at room temperature for 6 h [25]. The freeze-thaw cycle was carried out three times. The obtained cryogels were poured into plexiglass molds with the surface 14 × 14 cm and dried at 37 °C for 24 h. After drying, films were cut into pieces of 2 × 3 cm, which corresponded to the POS dose of 100 mg/film.

### 3.3. Films Evaluation

#### 3.3.1. Scanning Electron Microscopy (SEM)

To characterize the morphology of the prepared films, a scanning electron microscope (SEM) (Inspect™S50, FEI Company, Hillsboro, OR, USA) was utilized. Samples were sputter-coated with gold before imaging with a 5 nm thick layer of gold to make their surfaces conductive and were imaged at an accelerating voltage of 5 kV with an 8 mm work distance.

#### 3.3.2. Surface pH

To measure the surface pH of films, they were kept in a Petri dish containing 10 mL of distilled water for 10 min to swell. After swelling, the surface pH was measured by using an Orion 3 Star pH-meter glass electrode (Thermo Scientific, Waltham, MA, USA). pH probe was in contact with the surface of each film and was allowed to equilibrate for 1 min.

#### 3.3.3. Film Thickness

The thickness of each film was measured using a thickness gauge (Mitutoyo, Kawasaki, Japan). The measurement of the thickness of each film at six different locations (two in the middle part and four corners) was performed. For each formulation, three randomly selected films were used.

#### 3.3.4. Moisture Content

Content moisture was assessed using moisture analyzer balance (Radwag WSP 50SX, Radom, Poland).

#### 3.3.5. Disintegration Time

To examine the disintegration time of all prepared films, the conventional apparatus (Erweka ED-2L, Heusenstamm, Germany) and Petri dish were applied. Disintegration time in conventional apparatus was performed according to European Pharmacopoeia [13] with 700 mL of simulated saliva solution (SSS, pH 6.8) as a disintegration medium. To determine disintegration time using the Petri dish, films were carefully put in the center of the Petri dish (a diameter of 7 cm) containing 7 mL of SSS [48].

#### 3.3.6. Mechanical Properties

The mechanical properties expressed by tensile strength (TS), percent of elongation (E%), and Young’s modulus (E) were examined using Texture Analyzer TA.XT. Plus (Stable Microsystems, Godalming, UK). Based on the preliminary tests, the experimental parameters of the process were set as follows: pre-test speed 1 mm/s, test speed 1 mm/s, post-test speed 1 mm/s, distance 3 mm, strain 10%, trigger force 0.001 N, break sensitivity 0.107 N. TS was calculated by using formula, where applied stress (F) was dividing per area (A).
TS = F/A(1)
E% was calculated by the following equation:E% = [(L − L_0_)/L_0_] × 100(2)
where L_0_–length of the film prior to the experiment, and L–length of the films after the elongation.

Young’s modulus (E) is the parameter measured for the determination of film stiffness. It is defined by applied stress over strain ratio in the elastic deformation zone. Young’s modulus was calculated using the following equation [49]:E = (F/A)/(L − L_0_/L)(3)
where F–applied stress, A–film area, L_0_–length of the film prior to the experiment, and L–length of the films after the elongation.

Folding endurance was assessed by repeatedly folding each film at the same point until breaking occurred. The number of times a film was folded without breaking was expressed as folding endurance [48].

#### 3.3.7. Swelling Properties

Film swelling properties were expressed as a swelling index (SI). Films of definite size (2 × 3 cm) were weighed (W_1_) and placed on a Petri dish with 20 mL of SSS (pH 6.8) or distilled water. At the time intervals (5, 10, 15, 30, 60, 120, 180, 240 min), films were removed, and the excess of moisture was eliminated. Finally, films were weighed (W_2_), and SI was calculated based on the following formula [50]:SI = (W_2_ – W_1_)/W_1_(4)
where W_1_ is the film initial weight, and W_2_ is the film weight after swelling.

#### 3.3.8. Mucoadhesiveness

##### Ex Vivo Mucoadhesive Properties

The evaluation of mucoadhesiveness was performed using TA.XT.Plus Texture Analyzer (Stable Micro Systems, Godalming, UK) and bovine buccal mucosa as a natural adhesive model. Experimental parameters of the process were chosen during preliminary tests and set as follows: pretest speed 0.5 mm/s, test speed 0.1 m/s, contact time 180 s, post-test 0.1 mm/s, applied force 1 N. The mucoadhesive properties were determined as the maximum detachment force (F_max_) and the work of mucoadhesion (W_ad_).

##### Ex Vivo Residence Time

The residence time was evaluated by the adhesion test known as the “wash off” method. It was performed using self-constructed apparatus (by modifying USP disintegration tester), where plexiglass cylinder (6 cm diameter, weight 280 g) moving up and down was vertically fixed. The segments of bovine buccal mucosa (5 × 3 cm) were glued to the internal side of a beaker above the volume of 700 mL of SSS (pH 6.8) at 37 ± 0.5 °C. Films were put in contact with the mucosal membrane and immersed completely in the medium. The time required for the entire detachment of film from the mucosa was noted [51].

##### In Vivo Mucoadhesion Time

In vivo mucoadhesion time was carried out using placebo formulations P1–P4. Consent from the Institutional Ethical Committee according to the rules of GCP/Guidelines for Good Clinical Practice (number R-I-002/293/2018) was taken before carrying out the experiment. Permission from all the volunteers was also received. Healthy volunteers applied films to the inside of the cheek pocket, and the retention time was registered.

#### 3.3.9. Drug Content Uniformity

Films were placed in **graduated flasks** with 30 mL SSS (pH 6.8) at a rotating speed of 50 rpm at 37 ± 0.5 °C. After 24 h of agitation in a water bath, 20 mL methanol was added. The obtained solutions were filtrated and then analyzed by the HPLC method (as described in point 3.3.10. HPLC assay).

#### 3.3.10. High-Performance Liquid Chromatography (HPLC) Assay

POS concentration was studied by the HPLC method using an Agilent Technologies 1200 system (Agilent, Waldbronn, Germany) and a Poroshell^®^ 120 EC-C18 2.7 μM ODS 4.6 × 150 mm, 2.7 μm column (Agilent, Waldbronn, Germany). As the mobile phase, acetonitrile: methanol: water (60:20:20, v/v) with a flow rate of 0.5 mL/min was applied [52]. The UV detection was operated with a wavelength of 240 nm. The retention time of POS was observed at 5.5 min. The standard calibration curve was linear over the range of 1–100 μg/mL with the correlation coefficient (R^2^) 0.999.

#### 3.3.11. In Vitro Drug Release

For the in vitro POS dissolution test, apparatus type II (Erweka Dissolution Tester Type DT 600HH, Heusenstamm, Germany) was used [13]. Films were fixed using cyanoacrylate glue to the central shaft of the paddle. As a dissolution medium, 700 mL of SSS pH 6.8 with 1% sodium dodecyl sulfate (SDS) to obtain sink conditions was applied. The release study was performed at 37 ± 0.5 °C with a rotation speed of 100 rpm. POS concentration in the release medium was examined by the HPLC technique (as described in point 3.3.10. HPLC assay).

#### 3.3.12. Drug Release Mechanisms

POS-release kinetic was evaluated based on five mathematical models: zero-order, first-order, Higuchi, Korsmeyer–Peppas, and Hixson–Crowell [17,33].

#### 3.3.13. Differential Scanning Calorimetry (DSC)

DSC analysis of ALG, OLG, pure POS, placebo (P1, P2, P4), and POS-loaded formulations (F1, F2, F4) was performed using an automatic thermal analyzer system (DSC TEQ2000, TA Instruments, New Castle, DE, USA). The individual sample was precisely weighed (5 mg) and placed in a sealed aluminum pan. Temperature calibrations were performed using indium and zinc as standard. Samples were heated from 25 °C to 300 °C at the scanning rate of 10 °C/min under a nitrogen flow of 20 mL/min.

#### 3.3.14. Antifungal Activity

The antifungal activity of the films against yeast cultures *Candida parapsilosis* ATCC^®^ 22019, *Candida krusei* ATCC^®^ 6528, and *Candida albicans* ATCC^®^ 10231 was tested by the plate diffusion method according to the Clinical and Laboratory Standards Institute (CLSI) [43]. The POS, dissolved in DMSO, ALG, and OLG powders, was used as controls. The Sabouraud’s dextrose agar was used for susceptibility testing. The initial density of *Candida* was approximately 2–5 × 10^6^ colony forming units (CFU)/mL. The inoculum of fungi (with a density of 0.5 in McFarland scale) was prepared in sterile 0.9% NaCl solution in a final density of 5 × 10^4^ CFU/mL. The optical density of inoculum was measured spectrophotometrically at 600 nm using Genesys 10S UV-Vis spectrophotometer (Thermo Scientific, Madison, WI, USA) [53]. Then, Petri dishes containing Sabouraud’s dextrose agar were seeded with 50 µL of the *Candida* inoculum. After 15 min, rings of all formulations with 5 mm diameter were placed on the surface of the inoculated agar plates. In the next step, 5 mm diameter wells were cut in the agar basis, and 50 μL of POS solution in DMSO (corresponding to 3 mg of POS) was placed in a well. The plates were incubated at 37 ± 0.1 °C for 24 and 48 h. Then, the inhibition zones (mm) were measured with a caliper (Mitutoyo, Kawasaki, Japan) around each well with an accuracy of 0.1 mm.

#### 3.3.15. Statistical Analysis

Results were analyzed using Statistica 10.0 software (StatSoft, Tulsa, OK, USA). Quantity variables were expressed as the mean and standard deviation. Statistical analysis was performed using one-way analysis of variance (ANOVA) or Kruskal–Wallis test. Differences between groups were considered to be significant at *p* < 0.05.

## 4. Conclusions

In this study, POS-containing mucoadhesive films, based on ALG and OLG for buccal delivery, were designed by a new freeze-thaw method. The designed formulations showed a high degree of swelling, good mucoadhesive properties, and prolonged up to 5 h POS-release. The addition of 1% OLG to 1% ALG led to obtaining the optimal formulation (F2) and significantly improved the mechanical properties (films were soft and flexible). OLG presence in the formulation improved the mechanical properties evaluated as tensile strength, percent of elongation, and Young’s modulus. In the mucoadhesion test, the increase in W_ad_ value after OLG addition was observed. Importantly, it was shown that both ALG and OLG tested in a powder form, and placebo film formulations exhibited antifungal activity expressed as growth inhibition zones for *Candida spp.* strains. The obtained data has opened the way for future research for developing OLG-based dosage forms, which might increase the activity of antifungal drugs.

## Figures and Tables

**Figure 1 marinedrugs-17-00692-f001:**
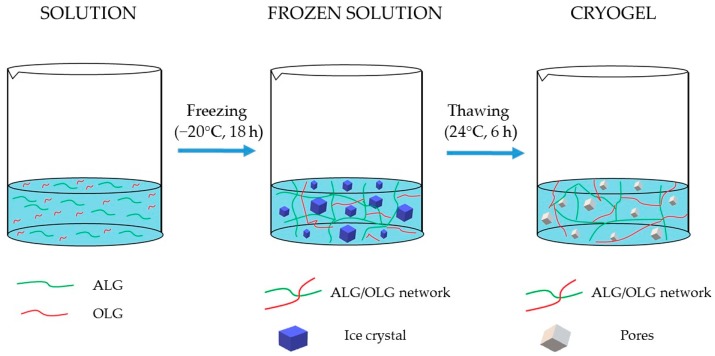
Scheme of ALG (sodium alginate) cryogels preparation by the freeze-thaw method.

**Figure 2 marinedrugs-17-00692-f002:**
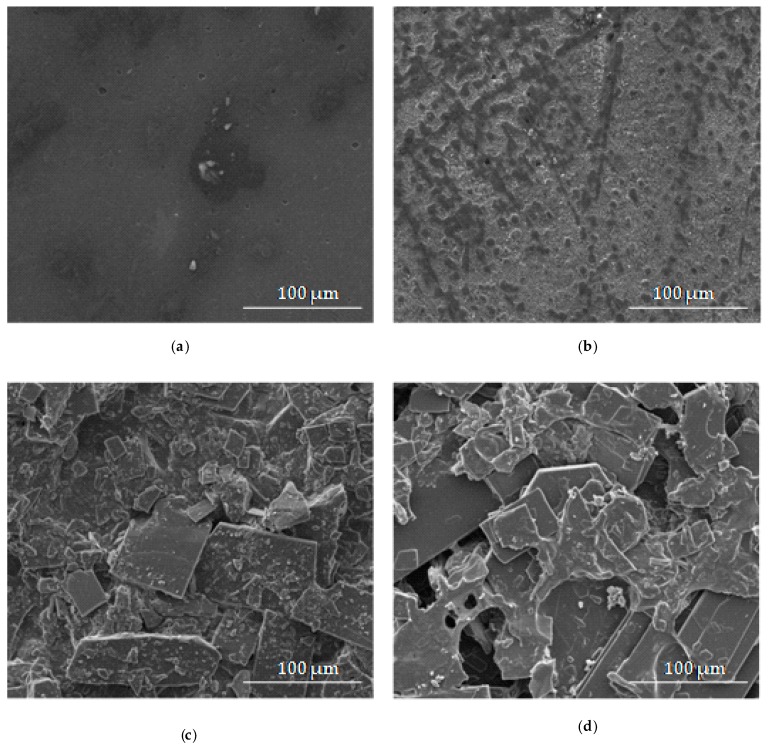
The surface morphology of film formulation (**a**) P1, (**b**) P4, (**c**) F1, and (**d**) F4 under magnification ×1000.

**Figure 3 marinedrugs-17-00692-f003:**
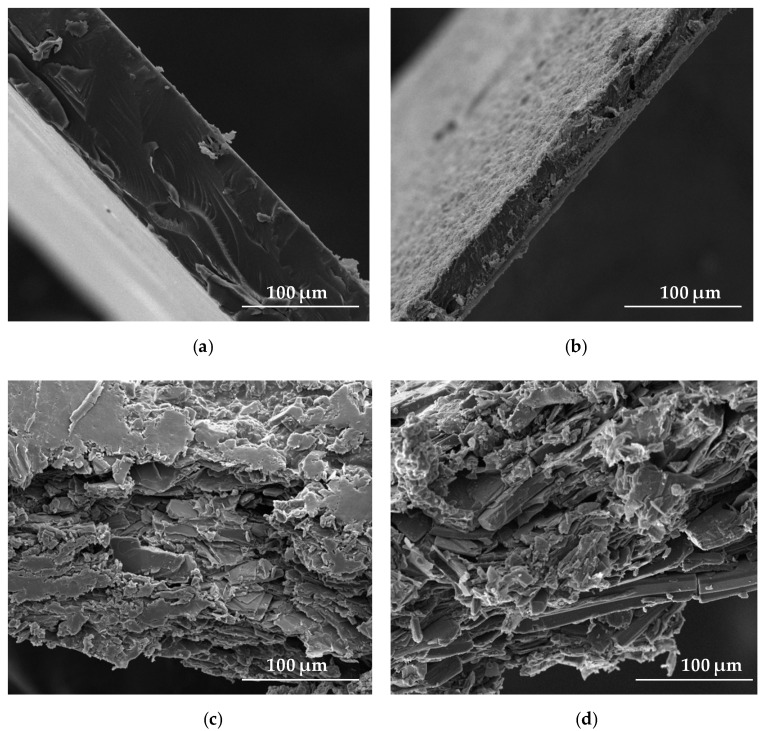
The cross-section of films formulation (**a**) P1, (**b**) P4, (**c**) F1, and (**d**) F4 under magnification ×1000.

**Figure 4 marinedrugs-17-00692-f004:**
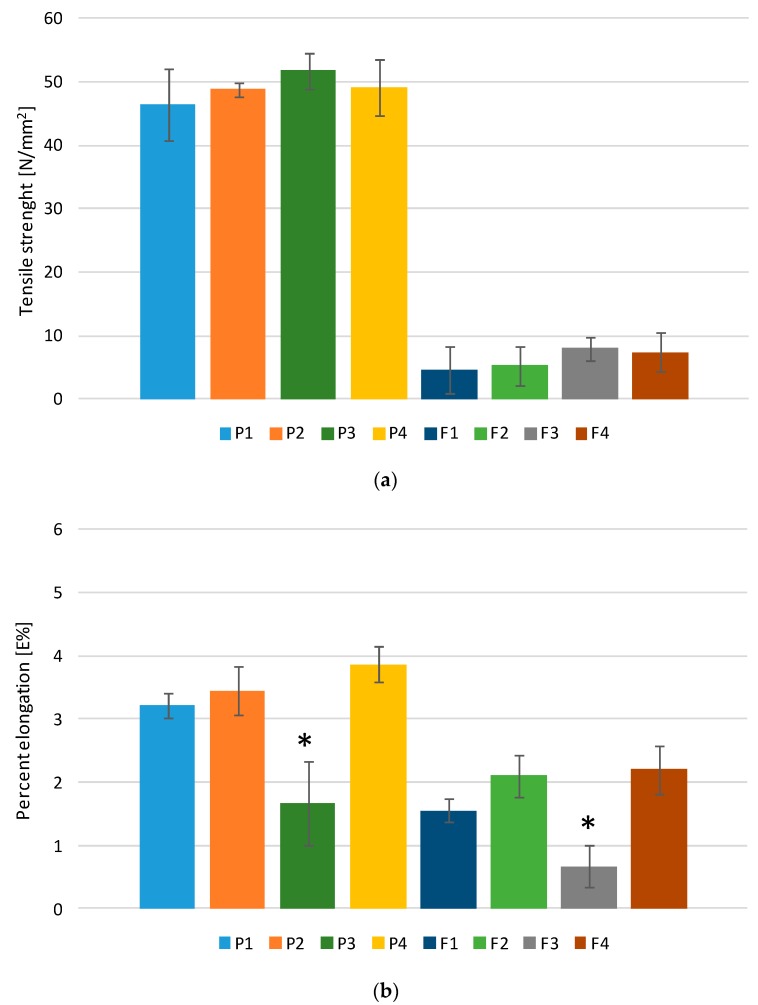
The mechanical properties of films with placebo (P1–P4) and POS-loaded formulations (F1–F4) expressed as (**a**) tensile strength, (**b**) percent at elongation, and (**c**) Young’s modulus. * *p* < 0.05 *versus* P1 or F1.

**Figure 5 marinedrugs-17-00692-f005:**
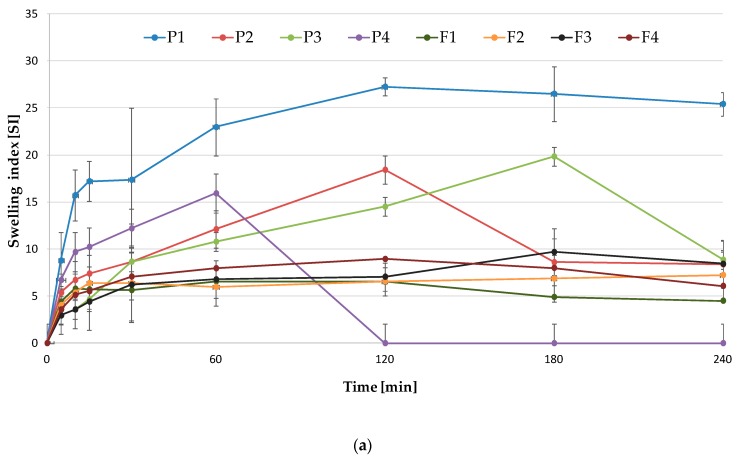
Swelling index (SI) of placebo (P1–P4) and POS-loaded (F1–F4) films in (**a**) simulated saliva solution (SSS) and (**b**) water.

**Figure 6 marinedrugs-17-00692-f006:**
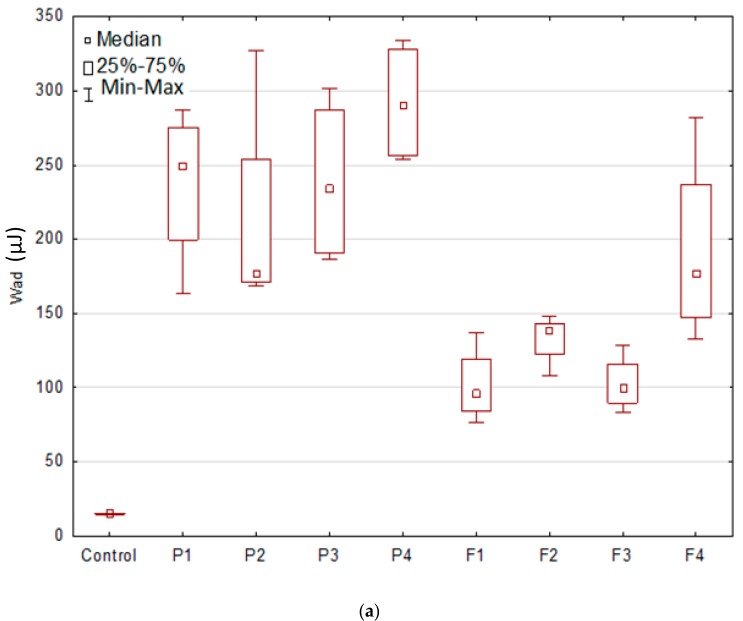
Mucoadhesive characteristics: (**a**) maximum detachment force (F_max_) and (**b**) work of adhesion (W_ad_) of placebo films (P1–P4), POS-loaded formulations (F1–F4), and cellulose paper (applied as control) (median; *n* = 6).

**Figure 7 marinedrugs-17-00692-f007:**
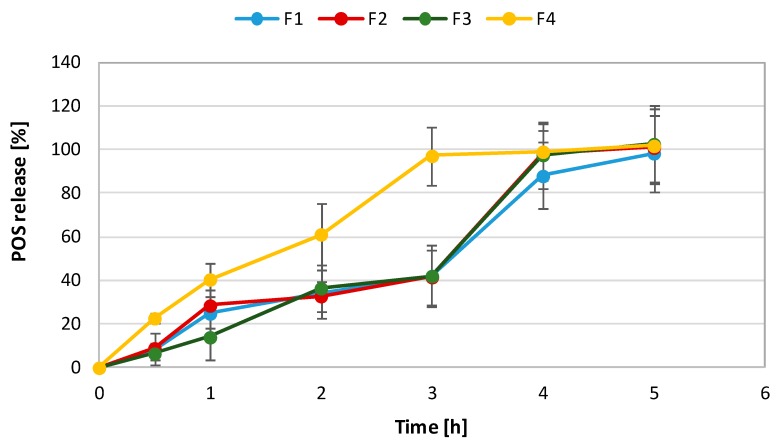
POS (posaconazole) dissolution from formulations F1–F4.

**Figure 8 marinedrugs-17-00692-f008:**
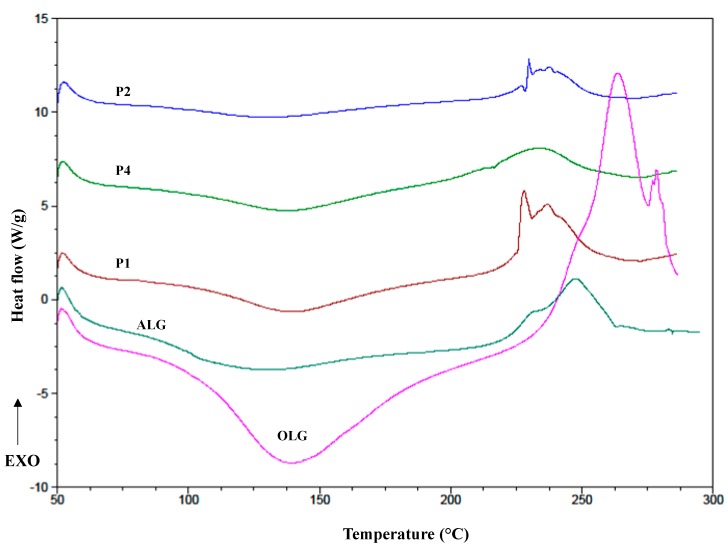
DSC (differential scanning calorimetry) thermograms of pure sodium alginate (ALG), alginate oligosaccharides (OLG), and placebo film formulations P1, P2, P4.

**Figure 9 marinedrugs-17-00692-f009:**
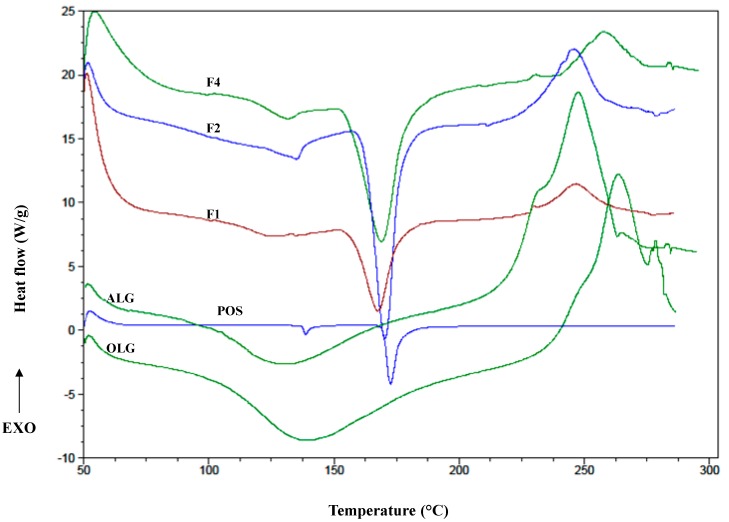
DSC thermograms of pure sodium alginate (ALG), alginate oligosaccharides (OLG) pure posaconazole (POS), and POS-loaded formulations F1, F2, F4.

**Figure 10 marinedrugs-17-00692-f010:**
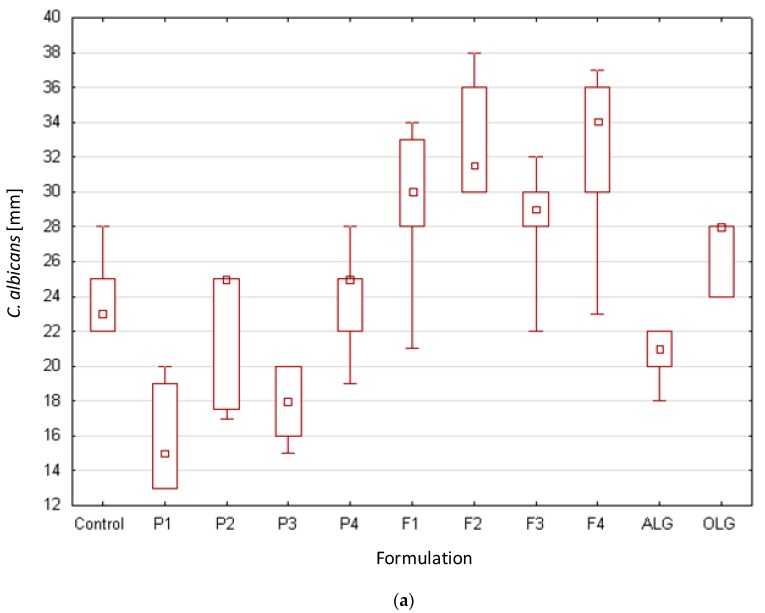
Box-plot graphs presenting antifungal activity of placebo film formulations (P1–P4) and POS-loaded films (F1–F4); reference standard (POS/DMSO) against standard strains of *Candida* spp. (**a**) *Candida albicans*, (**b**) *Candida*
*krusei,* and (**c**) *Candida parapsilosis* (*n* = 3).

**Table 1 marinedrugs-17-00692-t001:** Viscosity values of performed solutions and cryogels.

Formulation	Viscosity (mPa∙s)^*^
**Solutions**	
P1_sol_	3803.3 ± 57.7
P2_sol_	4028.4 ± 112.9
P3_sol_	48651.0 ± 114.3
P4_sol_	4291.0 ± 125.0
**Cryogels**	
P1_gel_	5473.8 ± 290.6
P2_gel_	6685.0 ± 290.2
P3_gel_	96610.0 ± 2737.1
P4_gel_	7679.0 ± 299.6

* *n* = 3.

**Table 2 marinedrugs-17-00692-t002:** Composition of placebo and POS-loaded formulations.

Formulation	ALG (g)	OLG (g)	GLY (g)	POS (g)	Purified Water (up to; g)
P1	2	-	0.6	-	100
P2	1	1	0.6	-	100
P3	2	2	0.6	-	100
P4	-	2	0.6	-	100
F1	2	-	0.6	3.27	100
F2	1	1	0.6	3.27	100
F3	2	2	0.6	3.27	100
F4	-	2	0.6	3.27	100

**Table 3 marinedrugs-17-00692-t003:** The characteristics of freeze-thaw prepared films: placebo and POS-loaded formulations.

Formulation	Thickness (μm)	Surface pH	Disintegration Time (min)		Moisture Content (%)	Drug Content (mg)
			Conventional Apparatus	on Petri Dish		
P1	52.0 ± 5.4	7.1 ± 0.1	3.4 ± 0.3	>240	8.8 ± 3.9	-
P2	54.7 ± 8.2	7.0 ± 0.1	3.2 ± 0.1	120 ± 5	9.1 ± 1.9	-
P3	76.7 ± 9.4	6.9 ± 0.1	6.5 ± 0.1	120 ± 5	9.4 ± 0.5	-
P4	42.6 ± 5.7	6.9 ± 0.1	2.2 ± 0.1	120 ± 5	7.3 ± 1.6	-
F1	213.8 ± 6.9	7.3 ± 0.3	16.1 ± 0.1	>240	7.2 ± 1.2	95.4 ± 0.1
F2	260.3 ± 8.6	7.2 ± 0.1	16.4 ± 0.1	>240	5.3 ± 1.5	101.8 ± 9.2
F3	287.0 ± 4.4	6.7 ± 0.1	19.2 ± 0.2	>240	9.7 ± 0.2	103.8 ± 6.2
F4	220.0 ± 8.1	6.7 ± 0.2	15.7 ± 0.5	>240	9.6 ± 0.4	103.1 ± 6.5

**Table 4 marinedrugs-17-00692-t004:** Ex vivo and in vivo residence time of designed mucoadhesive films.

Formulation	Ex Vivo Residence Time (min)	In Vivo Residence Time (min)
P1	133.33 ± 10.41	129.67 ± 10.02
P2	39.00 ± 5.29	50.33 ± 10.22
P3	111.33 ± 6.03	153.50 ± 1.80
P4	21.00 ± 3.61	54.17 ± 3.25
F1	205.00 ± 10.00	–*
F2	117.67 ± 7.51	–*
F3	180.67 ± 14.98	–*
F4	111.33 ± 4.04	–*

* not tested.

**Table 5 marinedrugs-17-00692-t005:** Models of POS-release from designed films.

Formu-Lation	Zero Order Kinetics		First Order Kinetics		Higuchi Model		Korsmeyer–Peppas Model			Hixson–Crowell Model	
	R^2^	K	R^2^	K	R^2^	K	R^2^	K	*n*	R^2^	K
F1	0.94	19.98	0.80	0.81	0.89	57.81	0.93	0.48	0.65	0.87	5.18
F2	0.89	20.88	0.64	0.94	0.85	60.29	0.90	0.48	0.63	0.82	5.76
F3	0.94	23.68	0.67	0.89	0.89	68.39	0.97	0.49	0.80	0.80	6.11
F4	0.94	22.39	0.84	1.27	0.97	67.29	0.95	0.50	0.47	0.94	1.60

R^2^: correlation coefficient, K: release constant, and *n*: the release exponent.

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
