# Peer review of "Alginate Oligosaccharides Affect Mechanical Properties and Antifungal Activity of Alginate Buccal Films with Posaconazole"

_marinedrugs, 2019, doi:10.3390/md17120692_

Round 1
Reviewer 1 Report
The manuscript entitled ”Alginate Oligosaccharides Affect Mechanical 2 Properties and Antifungal Activity of Alginate Buccal 3 Films with Posaconazole” can be published in Marine Drugs after grammar and spell-check. There are some errors in the text.
Author Response
RESPONSE TO THE REVIEWER # 1
We thank the Reviewer for the revision and the opinion about our manuscript.

Reviewer 2 Report
The article entitled "Alginate Oligosaccharides Affect Mechanical Properties and Antifungal Activity of Alginate Buccal Films with Posaconazole" is scientifically well explained research article and provide useful information’s to the readers.
However, I have several recommendations to this article before publish in marine drugs.
The authors need to discuss in details about Candida spp in the introduction section.
Is there any specific reason to select only Candida spp for this study against alginate buccal films?
Alginate oligosaccharides are principle component use in the present study. The physical properties and bioactivities are depending on the source of alginates. However, the authors not provide extraction method of extraction since they purchased those from Xi'an Haoxuan Bio-tech Co., Ltd. And Sigma. Thus, I’m recommend to provide source of alginate and cat numbers of alginate samples use for this study (section 3.1 Materials)
Figure 10: Details of y axis insufficient.
Unify the scientific name writing style throughout the manuscript. EX. L 580 and 582, 567, 579, 582
Authors used large number of citations (69). This is too much for regular research article. Please try to reduce number of citations mentions in the text.
In some figures authors do not provide significant details (EX-figure 4).
Author Response
RESPONSE TO THE REVIEWER # 2
Taking into account the comments raised by the Reviewer, the sentences about Candida ssp. and their selection for the study (page 1, lines 34-42) were added as follows:
“Candida is a genus of yeats belonging to the normal, commensal microbiota of gastrointestinal tract, vagina, skin and oral cavity or others mucosal surfaces in healthy humans. In case of patients critically ill, with immunodeficiency or AIDS, patients that use total parenteral nutrition, after transplantation and treated by cytotoxic chemotherapies or broad-spectrum antibiotics, Candida spp. might contribute to systemic infection. Although the genus Candida consists of many various species, predominantly etiological agents of human infection are Candida albicans, Candida parapsilosis, Candida tropicalis and Candida krusei. In addition, Candida albicans causes oral candidiasis – one of the most common opportunistic fungal infections, where local drug dosage forms are effective, especially buccal films.”
According to the suggestion of the Reviewer, the source and CAS number of alginate and alginate oligosaccharides were added to the section 3.1. Materials. Taking into account the comments raised by the Reviewer, in the Figure 10 details of y axis were added. According to the suggestion of the Reviewer, name writing style was corrected. Taking into account the comments raised by the Reviewer, in the Section References number of citations was reduced. According to the suggestion of the Reviewer, the significant details in the Figure 4 were added.
We thank the Reviewer for the revision and these insightful suggestions. We find Reviewer suggestion to be very helpful and constructive and the corresponding revisions have strengthened the paper in multiple ways. The questions are of interest and will be addressed in our current and future work.

Reviewer 3 Report
The aim of this work was to make alginate films for buccal delivery of the antifungal agent posaconazole and study the release profile, mechanical and mucosaddhesive properties as well as antifungal activity.
The films were produced by drying alginate cryogels, obtained from freeze-thawing cycle.
The characterization involved tensile strength and stress strain measurements, swelling measurements and posaconazole release profiles.
The paper is in general well written, but the descripton of the samples produced could be made clearer.
Conclusions are sound, allthough some controll experiments should be done.
These are suggested under specific comments but to sum up:
Rheological Characterization (G*, G** and phase angle) of alginate solutions before and after freeze-thawing cycles.
characterization of one of the alginate- posaconazole films without freeze thawing method. (for example P1 and F1)
Characterization of films with different loading of posaconazole (one sample).
In addition, it could be considered alternative methods for mixing posacoazole into the alginate cryogel before drying
Specific comments
Line 58. Why abreviation of oligosaccharides to OLG?
Line 68. Ref 14 and 15 study the effect of alginate oligomers rich in guluronic acid. One of the main motivations in these works was the hypothesis that G-oligomers will disrupt biofilms by removal of Ca2+ which binds selectively to G oligomers ( egg-box model). M oligomers will also have an affinity to Ca2+ due to it 's charge but binds weaker than G- oligomers.
Line 71-73. The concept is not very clear to me. Will the sample form a contineous network with aggregates acting as junction zones?
Will it also work for depolymerised samples (OLG).
As a rough estimate Mw = 2.3*103 will correspond to an average chain length of 5.8 assuming
Mw/Mn ≈ 2. Considering that the percistence length of alginate is around 12 nm and at least 3 junctions on each chain are needed to form a contineous network, OLG will not form a hydrogel in the ordinary sense. But again I am not certain how the structure will change after freeze-thawing.
Comment on introduction: Perhaps a few sentences on exactly how the film is administered for readers not familiar with this.
Line 90: What is meant by good elasticity? i.e how long is the linear region of a plot of stress vs. strain?
Line 98-99: Are the samples gelled if they are readily poured into moulds after freezeing-thawing?
Allthough the process had an clear effect on viscosity, it would be important to also measure the phase angle of the samples before and after freeze-thawing since this provides the simplest operational definition of a gel.
Table 1. Short sample description and concentrations. What is P2 and P3?
This is stated in table 5 but should be presented earlier.
Is the first four rows the alginate solutions before freeze-thawing ?
In that case it is not a hydrogel.
Could be a different code for samples with and without freeze-thawing
e.g P1sol and P1gel.
Section 2.1
It should be stated clearly which samples are hydrogels and (dried) films.
Line 113-117: It seems to me like (c) and (d) in figure 2 is crystaline and that the original polymer network is no longer present. How are the samples dried ? Phase transition may destroy the original structure. Can critical point drying be an alternative.
Line 188-189: "It was observed that films thickness increased significantly (p<0.05) with the increase in ALG concentration and POS presence"
The effect of POS is much larger than the effect of the different alginate preparations (table 2). How much of the sample did it constitute?
Line 194: It was perceived...
How was it measured?
Line 209: In table 2, the samples are listed. Since the freeze-thaw technique is central in this work, I miss negative controls. It would be natural to compare POS loaded films of alginate cryogels with
POS loaded films made from dried alginate solutions to evaluate the effect of the freeze-thaw process.
Line 212: In figure 2, the mechanical properties are compared between the different alginate formulations.
However the largest differences seem to be between placebo and POS loaded films, for example in 2 a).
I understand that it would be a large task to also do measurements as a function of POS concentration, but it would have been interesting to see the effect on at least one sample.
Line 357. in fig a)The decrease in swelling after 120, 180 and 240 minutes for P4, P2 and P3 respectively in figure 5 is explained by entering of calcium from the medium but it seems that there is no calcium in SSS (line 625).
Could reduced swelling of OLG added films also be explained by diffusion of short alginate oligomers from the film, thereby reducing the osmotic pressure?
in figure 5 b) It seems like the films are swelling the first 5 minutes and thereafter shrinking.
is W1 the initial weight of the dried film?
I think W1 should be the weight of the film immediately after hydration.
Line 592: Units on y axis ?
Line 631: What was the concentration of alginate, OLG and glycerol?
Line 648-649: is "hydrogel base" the alginate solution?
Would there be an other way of mixing POS with the alginate solution to ensure that it is homogeneously dispersed, e.g solving it first in an appropriate apolar solvent and use an homogenizer after addition?
The particle size of the added POS would potentially affect the release rate.
Author Response
RESPONSE TO THE REVIEWER # 3
We agree with the Reviewer that rheological characterization (G*, G** and phase angle) of alginate solutions before and after freeze-thawing cycles, characterization of the alginate-posaconazole films without freeze thawing method, characterization of films with different loading of posaconazole and alternative methods for mixing posaconazole into the alginate cryogel before drying are valuable issues that would enable to more accurately characterize the impact of the method of preparation, and polymers and drug substance effects on the properties of mucoadhesive buccal films. We thank the Reviewer for indicating the direction, which will be applied in next stages of our research. However, due to the fact that the goal of this manuscript was mainly to determine the effect of alginate oligosaccharides on antifungal activity, and because we received short time to respond, it is impossible to perform all the above mentioned studies. It should be emphasized that before conducting the experiments described in this manuscript, we prepared many various preliminary tests, which aimed to compare two methods of developing placebo films - solvent casting and freeze-thaw and impact of the applied method on films properties. It was observed that solvent casting and freeze thaw methods might be successfully used to prepare ALG films, which were characterized by thickness uniformity. Films formulated by freeze thaw method possessed significantly better mechanical, swelling and mucoadhesive properties. We thank the Reviewer for the suggestion of characterizing films with different methods of mixing the substance with the cryogel. These studies are the subject of our subsequent work related to the development of the composition of buccal films with posaconazole. In addition, it is planned to study the impact of ball milling process on the properties of posaconazole and its effect on film properties.
Response to the specific comments:
In the literature, different abbreviation of alginate oligosaccharides (e.g. AOS) can be found. In our study, the alginate oligosaccharides abbreviation was chosen by us in order to easier distinguish between the applied substances. We agree with Reviewer that References:
- Pritchard, M.F.; Powell, L.C.; Jack, A.A.; Powell, K.; Beck, K.; Florance, H.; Forton, J.; Rye, P.D.; Dessen, A.; Hill, K.E.; Thomas, D.W. A low-molecular-weight alginate oligosaccharide disrupts pseudomonal microcolony formation and enhances antibiotic effectiveness. Antimicrob. Agents Chemother. 2017, 61, pii: e00762-17. DOI: 10.1128/AAC.00762-17. )
- Tøndervik, A.; Sletta, H.; Klinkenberg, G.; Emanuel, C.; Powell, L.C.; Pritchard, M.F.; Khan, S.; Craine, K.M.; Onsøyen, E.; Rye, P.D.; Wright, C.; Thomas, D.W.; Hill, K.E. Alginate oligosaccharides inhibit fungal cell growth and potentiate the activity of antifungals against Candida and Aspergillus spp. PLoS One. 2014, 19, e112518. DOI: 10.1371/journal.pone.0112518.
study the effect of alginate oligomers rich in guluronic acid. Due to the small number of works devoted to the activity of alginate oligosaccharides and the lack of research on the antifungal activity of oligosaccharides composed predominantly of mannuronic acids, we cited those papers, which showed the antifungal activity of oligosaccharides and their effect on the activity of conventional antifungal drugs.
Taking into account comments raised by the Reviewer, the sentences:
“This technique is based on the solvent crystallization during freezing, which leads the polymers chains to connect into structures that became the junction zones of hydrogels after thawing”
was changed as follow:
“This technique is based on three phases: freezing of a monomeric or polymeric solution or zol, its storage in the frozen state and then – thawing. Solvent crystallization during freezing leads the polymer chains to connect into structures with the large amount of macropores. ALG gels obtained using freeze thaw method demonstrate a significant shrinkage that indicates a high degree of a balance between repulsive and attractive interactions which in the consequence leads to the creation of junction zones through van der Waals forces and between uncharged ALG hydrogen bonds chains in unfrozen liquid phase during thawing.”
We agree with the Reviewer that it is not known how OLG structure will change after freeze-thawing process and further studies are needed.
Taking into account the comments raised by the Reviewer, on the page 2, lines 53-56: the definition and explanation of application buccal films were added as follow: “Buccal films are mucoadhesive oral dosage forms that consist of hydrophilic polymer, active substance and other excipients. Films are applied on the inner membrane of the buccal mucosa, where after contact with the saliva they hydrate, adhere and dissolve drug that results in local or systemic effect.”
According to the suggestion of the Reviewer , the sentence:
“The obtained cryogels are characterized by soft, porous structure with good elasticity and high permeability for drug molecules”.
was changed as follow:
“The obtained cryogels are characterized by elasticity and soft, porous structure with high permeability for drug molecules.”
Elasticity is an ability of a deformed material to return to its original shape and size when the forces causing the deformation are removed. It is characterized by low value of Young's modulus, high values of tensile strength and percent of elongation [Semmling B., Nagel S., Sternberg K. et al.: Long-term stable hydrogels for biorelevant dissolution testing of drug-eluting stentsJournal of Pharmaceutical Technology & Drug Research 2013, 2, 19.; Brachkova M.I., Duarte A., Pinto J.F.: Alginate films containing viable Lactobacillus plantarum: preparation and in vitro evaluation. AAPS PharmSciTech. 2012, 13(2):357-63.]. Young’s modulus is described as a measure of the stiffness of a material and reflects the number and length of elastic segments. Therefore, cryogels obtained with the freeze-thaw technique are characterized by elasticity, and the length of the linear region of a stress vs. strain curve depends on the polymers, the preparation method and the additional ingredients.
Taking into account the comments raised by the Reviewer, measuring the phase angle of samples is planned. Based on the pre-formulation tests, the polymer concentrations were selected to enable transfer of the cryogels to the plexiglass moulds. Formulations prepared with higher polymer concentrations possessed higher viscosity values and were difficult to pour. Taking into account the comments raised by the Reviewer, in the Section 2.3.1 Table 2 with formulation composition was added. In the Table 1 formulations codes before and after freeze-thaw process were changed. In addition, according to the suggestion of the Reviewer, the nomenclature has been changed - the hydrogel has been replaced by term solution. In the Section 2.1 all samples are dried films – figure 2 presents the surface morphology of films formulation P1, P4, F1 and F4, but figure 3 shows cross-section of these films formulation. We agree with the Reviewer that films in the Figure 2 c and d possess crystalline structure and original polymer network is invisible. The samples were dried at 37 ± 1°C for 24 h using laboratory dryer. We thank the Reviewer for the suggestion of using critical point drying. Application of this method might obtain less artifacts on the surface of the tested samples and leave the original gel structure unchanged. The dose of posaconazole in the films was 100 mg and was selected based on the posaconazole content in commercially available tablets. Because the dose of substance was high, posaconazole constituted a large part of the film. Examination of different posaconazole doses and their effect on the pharmaceutical properties of films will be the next stage of our research. Determination of drug content uniformity was described in the point 3.3.9. Drug Content Uniformity. It was measured by placing films in a graduated flasks with 30 mL SSS (pH 6.8) at rotating speed of 50 rpm at 37 ± 0.5°C. After 24 h of agitation in a water bath, 20 mL of methanol was added. The obtained solutions were filtrated and then analyzed by the HPLC method (as described in the point 3.3.10. HPLC assay). We agree with the Reviewer, that very valuable is comparing POS-loaded films prepared by freeze-thaw method with POS-loaded films obtained by solvent casting method. However, we have conducted many preliminary tests, which aimed to compare two methods of developing placebo films - solvent casting and freeze-thaw techniques and their impact on films properties. It was observed that solvent casting and freeze thaw methods might be successfully used to prepare films, which were characterized by thickness uniformity. Films formulated by freeze thaw method possessed significantly better mechanical, swelling and mucoadhesive properties. We agree with the Reviewer, that the largest differences of mechanical properties were observed between placebo and POS loaded films. Determining the changes in mechanical properties as a function of POS concentration is a very interesting aspect, therefore such tests are planned to be carried out in the next stage of research on designed films. Taking into account the comments raised by the Reviewer, there was a mistake in the manuscript regarding explanation of decrease swelling index by entering into the films matrix of calcium ions from the medium, but simulated saliva solution does not contain calcium ions.
The sentences:
“Ca2+ ions present in the medium enter into the ALG matrix and undergo ion exchange with Na+ ions, which are attached to carboxylate groups of ALG M-blocks. It results in the relaxation of M chains and in the consequence – in the SI increase. When film structure is fully hydrated it begins to lose its structural integrity due to the disruption of “egg-box” cavities and ALG chains begin to disintegrate and dissolve. It is known that ALG with predominated M-block content is characterized by higher water absorption and it exchanges ions more easily in comparison to ALG related with a high number of G-blocks, which linked with Ca2+ ions lead to the formation of the “egg-box” conformation, not dissolving in water and possessing the ability to slower water uptake”
were changed as follow:
“Short chains of OLG oligomers present in the film matrix diffuse into the medium thereby reducing osmotic pressure. It results in the relaxation of polymer chains and in the consequence – in the SI increase. When film structure is fully hydrated it begins to lose its structural integrity due to the medium penetration into the matrix and ALG chains begin to disintegrate and dissolve. Prolonged swelling of ALG formulations is related with a high number of G-blocks, which form an integral matrix slowly dissolving in SSS and possessing the ability to slower water uptake. In addition, presence of Na+ ions in the medium leads to electrostatic repulsion and in the consequence causes the chains disintegration. In the gel matrix free counter-ions form an 'ionic' osmotic pressure between solvent and gel, which provides the slow solvent diffusion into the film polymer network.”
We agree with the Reviewer, that films placed in water (Figure 5b) are characterized by swelling at the first 5 minutes and then their surface is reduced as a result of dissolution of alginate and its oligosaccharides in the applied medium. Swelling index was calculated based on the methodology and mathematical equation available in the literature and the symbol W1 is the initial weight of dried film (Bahri-Najafi, R.; Tavakoli, N.; Senemar, M.; Peikanpour, M. Preparation and pharmaceutical evaluation of glibenclamide slow release mucoadhesive buccal film. Pharm. Sci. 2014, 9, 213–223.; Çelik, B. Risperidone mucoadhesive buccal tablets: formulation design, optimization and evaluation. Drug Des Devel Ther. 2017, 11, 3355–3365. DOI: 10.2147/DDDT.S150774; Szekalska, M.; Sosnowska, K.; Zakrzeska, A.; Kasacka, I.; Lewandowska, A.; Winnicka, K. The influence of chitosan cross-linking on the properties of alginate microparticles with metformin hydrochloride – in vitro and in vivo evaluation. Molecules 2017, 22, 182. DOI: 10.3390/molecules22010182; Kaur, K.; Naeem, M., Ali, A.; Rehman, N.U.; Nawaz, Z.; Akram, M.R.; Khan, J.A. Assessment of guar and xanthan gum based floating drug delivery system containing mefenamic acid. Acta Pol. Pharm. 2016, 5, 1287-1297.) Taking into account the comments raised by the Reviewer, the unit on the y axis in Figure 10 was added. According to the suggestion of the Reviewer, concentrations of ALG, OLG and glycerol was added and the sentence:
“Briefly, ALG solutions were obtained by dissolving proper amount of polymer in water with glycerol (used as plasticizer) and stirred using mechanical stirrer model DT 200 (Witko, Łódź, Poland) until homogenous mixture appeared. Then, into polymer solution OLG was added with continuously stirring.” was changed as follow:
“Briefly, ALG solutions were obtained by dissolving proper amount of polymer in water (1% w/w or 2% w/w) with 0.6% (w/w) glycerol (used as plasticizer) and they were stirred using mechanical stirrer model DT 200 (Witko, Łódź, Poland) until homogenous mixture appeared. Then, into polymer solution, OLG (1% w/w or 2% w/w) was added with continuously stirring. Formulation P1 was prepared by addition 0.6% (w/w) glycerol to the 2% (w/w) ALG solution, and formulation P4 – to 2% (w/w) OLG solution.”
According to the suggestion of the Reviewer, the sentence:
“Proper amount of POS was micronized in a mortar for 20 minutes and then mixed with the hydrogel base.”
was changed as follow:
“Proper amount of POS was micronized in a mortar for 20 minutes and then mixed with the ALG, OLG or ALG/OLG solutions”.
We agree with the Reviewer that the way of mixing POS with the alginate solution can affect properties of designed films, therefore the impact of using ball milling process will be examined and described in a due course.
We thank the Reviewer for the revision and these insightful suggestions. We find Reviewer suggestion to be very helpful and constructive and the corresponding revisions have strengthened the paper in multiple ways. The questions are of interest and will be addressed in our current and future work.

Round 2
Reviewer 2 Report
Overall revised version of “Alginate Oligosaccharides Affect Mechanical Properties and Antifungal Activity of Alginate Buccal Films with Posaconazole”, is addressed the reviewer questions and suggestions as expected. Therefore, I’m suggesting to consider this manuscript to publish in Marine Drugs
Reviewer 3 Report
Comments: I would thank the authors for addressing most of my questions. I understand that it is not possible to do many experiments with the given time limit. It is the responsibility of the journal to provide enough time for the authors to address the questions asked by the reviewers. However, since the authors already have done tests comparing the effect of solvent casting and freeze-thawing on film properties, it would be good to include these either in the result section or in supporting information. Line 82: This question is very complex and I do not expect the authors to give a complete answer. Perhaps it is best to leave it until one has more details. 10. Ok, perhaps I misunderstood. I thought the drug uniformity in a single film was measured, but it is drug content in several films that is compared? 11. Could the results from some of these tests be addd to the paper? Some minor text editing can be done. Otherwise the manuscript can be accepted in the present form.